# Safety and parasite clearance of artemisinin-resistant *Plasmodium falciparum* infection: A pilot and a randomised volunteer infection study in Australia

**Rebecca E. Watts**[1], **Anand Odedra**[1], **Louise Marquart**[1], **Lachlan Webb**[1], **Azrin N. Abd-Rahman**[1], **Laura Cascales**[1], **Stephan Chalon**[2], **Maria Rebelo**[1], **Zuleima Pava**[1], **Katharine A. Collins**[1], **Cielo Pasay**[1], **Nanhua Chen**[3], **Christopher L. Peatey**[3], **Jörg J. Möhrle**[2], **James S. McCarthy**[1]*

**1** QIMR Berghofer Medical Research Institute, Brisbane, Australia, **2** Medicines for Malaria Venture, Geneva, Switzerland, **3** Australian Army Malaria Institute, Brisbane, Australia

* j.mccarthy@uq.edu.au

## Abstract

### Background

Artemisinin resistance is threatening malaria control. We aimed to develop and test a human model of artemisinin-resistant (ART-R) *Plasmodium falciparum* to evaluate the efficacy of drugs against ART-R malaria.

### Methods and findings

We conducted 2 sequential phase 1, single-centre, open-label clinical trials at Q-Pharm, Brisbane, Australia, using the induced blood-stage malaria (IBSM) model, whereby healthy participants are intravenously inoculated with blood-stage parasites. In a pilot study, participants were inoculated (Day 0) with approximately 2,800 viable *P. falciparum* ART-R parasites. In a comparative study, participants were randomised to receive approximately 2,800 viable *P. falciparum* ART-R (Day 0) or artemisinin-sensitive (ART-S) parasites (Day 1). In both studies, participants were administered a single approximately 2 mg/kg oral dose of artesunate (AS; Day 9). Primary outcomes were safety, ART-R parasite infectivity, and parasite clearance. In the pilot study, 2 participants were enrolled between April 27, 2017, and September 12, 2017, and included in final analyses (males $n = 2$ [100%], mean age = 26 years [range, 23–28 years]). In the comparative study, 25 participants were enrolled between October 26, 2017, and October 18, 2018, of whom 22 were inoculated and included in final analyses (ART-R infected participants: males $n = 7$ [53.8%], median age = 22 years [range, 18–40 years]; ART-S infected participants: males $n = 5$ [55.6%], median age = 28 years [range, 22–35 years]). In both studies, all participants inoculated with ART-R parasites became parasitaemic. A total of 36 adverse events were reported in the pilot study and 277 in the comparative study. Common adverse events in both studies included headache, pyrexia, myalgia, nausea, and chills; none were serious. Seven participants

**Data Availability Statement:** All raw data obtained in the two clinical trials described are included in

S1 Appendix (pilot study) and S2 Appendix (comparative study).

**Funding:** This study was funded by the Bill & Melinda Gates Foundation (OPP1111147; https:// www.gatesfoundation.org). Medicines for Malaria Venture was the recipient of this grant; SC and JJM are employees of Medicines for Malaria Venture. The funders had no role in study design, data collection and analysis, decision to publish, or preparation of the manuscript.

**Competing interests:** I have read the journal's policy and the authors of this manuscript have the following competing interests: REW, AO, LM, LW, ANA, LC, MR, ZP, KAC, CP, and JSM are employees of the study sponsor QIMR Berghofer Medical Research Institute; SC and JJM are employees of Medicines for Malaria Venture which provided funding for the study.

**Abbreviations:** A/P, atovaquone/proguanil; ALT, alanine aminotransferase; ART-R, artemisinin-resistant; ART-S, artemisinin-sensitive; AS, artesunate; BMI, body mass index; CI, confidence interval; DHA, dihydroartemisinin; IBSM, induced blood-stage malaria; IV, intravenously; LLN, lower limit of normal; $\log_{10}PRR_{48}$, parasite reduction ratio per 48 hours in the logarithmic base 10 scale; NA, not applicable; NR, not recorded; PQ, primaquine; PQP, piperaquine phosphate; qPCR, quantitative PCR; qRT-PCR, reverse transcription qPCR; QTcF, QT interval corrected using Fridericia's formula; ULN, upper limit of normal; VIS, volunteer infection studies.

experienced transient severe falls in white cell counts and/or elevations in liver transaminase levels which were considered related to malaria. Additionally, 2 participants developed ventricular extrasystoles that were attributed to unmasking of a predisposition to benign fever-induced tachyarrhythmia. In the comparative study, parasite clearance half-life after AS was significantly longer for ART-R infected participants ($n$ = 13, 6.5 hours; 95% confidence interval [CI] 6.3–6.7 hours) compared with ART-S infected participants ($n$ = 9, 3.2 hours; 95% CI 3.0–3.3 hours; $p$ < 0.001). The main limitation of this study was that the ART-R and ART-S parasite strains did not share the same genetic background.

## Conclusions

We developed the first (to our knowledge) human model of ART-R malaria. The delayed clearance profile of ART-R parasites after AS aligns with field study observations. Although based on a relatively small sample size, results indicate that this model can be safely used to assess new drugs against ART-R *P. falciparum*.

## Trial registration

The studies were registered with the Australian New Zealand Clinical Trials Registry: ACTRN12617000244303 (https://www.anzctr.org.au/Trial/Registration/TrialReview.aspx?id=372357) and ACTRN12617001394336 (https://www.anzctr.org.au/Trial/Registration/TrialReview.aspx?id=373637).

## Author summary

### Why was this study done?

- Malaria resistance to artemisinin combination therapies is spreading in the Greater Mekong subregion; therefore, new antimalarial drugs are needed to control malaria.

- Malaria volunteer infection studies (VIS), in which healthy volunteers are infected with malaria parasites, have been used to test antimalarial drugs in development.

- We sought to develop a human model of artemisinin-resistant (ART-R) malaria that could be used to evaluate the efficacy of antimalarial drugs against ART-R malaria.

### What did the researchers do and find?

- We conducted 2 malaria VIS. In a pilot study, 2 healthy participants were infected with ART-R parasites. In a comparative study, participants were randomised to be infected with either ART-R (13 participants) or artemisinin-sensitive (ART-S; 9 participants) parasites. In both studies, participants were given a single dose of an artemisinin derivative (artesunate[AS]) 8 or 9 days after infection.

- Malaria was well tolerated in the pilot study (36 adverse events reported) and the comparative study (277 adverse events reported); no serious adverse events were reported in the studies. Common adverse events included headache, pyrexia, myalgia, nausea, and chills.

- In the comparative study, parasites took significantly longer to clear from the blood of participants after AS administration for participants infected with ART-R parasites (6.5 hours) compared with participants infected with ART-S parasites (3.2 hours).

### What do these findings mean?

- We have developed the first (to our knowledge) human model of ART-R malaria, the results of which, although based on a relatively small sample size, indicate potential to be safely used to assess the efficacy of new drugs against ART-R parasites.

- The longer time to clear ART-R parasites from the blood of participants after AS administration is comparable with the observations from studies conducted in the Greater Mekong subregion, which suggests that the findings from this human model of ART-R malaria are relevant for clinical malarial patients.

## Introduction

Resistance to artemisinin-based combination therapies—the first-line treatment for malaria—is spreading in the Greater Mekong subregion [1] and is threatening malaria control. Artemisinin resistance is characterised by a delayed clearance phenotype, with a parasite clearance half-life >5 hours [2]. This drug-resistant phenotype is linked to a number of specific nonsynonymous mutations in the *Plasmodium falciparum kelch13* gene [3,4]. New drugs to treat malaria infection with artemisinin-resistant (ART-R) parasites are needed.

The development of antimalarial drugs has been accelerated by volunteer infection studies (VIS), in which healthy volunteers are experimentally infected with malaria parasites to evaluate the efficacy of candidate antimalarial drugs [5]. VIS allow early accrual of efficacy and dose-response data, thereby minimising the risk of failure when conducting phase 2 studies in patients with clinical malaria. The induced blood-stage malaria (IBSM) model, a type of VIS in which volunteers are inoculated with *Plasmodium*-infected erythrocytes [6–9], is particularly suitable to evaluate drug efficacy against blood-stage infection, the parasite stage that causes malaria morbidity and mortality. To date, the IBSM model has only been used with artemisinin-sensitive (ART-S) parasites, mainly with 3D7, the reference strain of *P. falciparum*.

Here, we present the results from 2 clinical trials. First, we conducted a pilot study in which we inoculated participants with ART-R *P. falciparum* parasites harbouring the *kelch13* gene mutation R539T (K13$^{R539T}$ strain) [4] to determine the safety, tolerability, and clearance of infection with ART-R parasites. Then, in a comparative study, we randomised participants to be inoculated with ART-R (K13$^{R539T}$ strain) or ART-S (3D7 strain) parasites and compared the clearance profiles of these 2 parasite strains after administration of a single oral dose of artesunate (AS). We used single-dose AS monotherapy rather than a longer course of AS or artemisinin-based combination therapy because this study design provided the best opportunity to investigate the pharmacodynamic effect of AS alone on the clearance of ART-R and ART-S parasites without the confounding effect of a partner drug. Furthermore, we had previously observed that single-dose drug administration with subcurative intention is a very effective design for characterising the pharmacodynamic effect of an antimalarial drug [7]. The aim of this study was to evaluate the safety and infectivity of ART-R parasites and to compare the parasitological response of the ART-R and ART-S parasites to AS in healthy participants inoculated in the IBSM model. We hypothesised that AS would clear parasitaemia from

participants infected with ART-R parasites at a slower rate compared to participants infected with ART-S parasites, as observed in field studies [2].

## Methods

### Study design

We conducted 2 consecutive phase 1, single-centre, open-label studies: a pilot study and a randomised study. In both studies, healthy participants were inoculated with ART-R parasites using the IBSM model. In the pilot study, 2 participants were inoculated with ART-R parasites with a 3-week interval in between. In the comparative study, participants were randomised to receive ART-R or ART-S parasites. The studies were conducted at Q-Pharm Pty Ltd, Brisbane, Australia, and approved by both the QIMR Berghofer and Australian Red Cross Blood Service Human Research Ethics Committees. The studies were registered with the Australian New Zealand Clinical Trials Registry: ACTRN12617000244303 and ACTRN12617001394336. Data were analysed according to the clinical trial protocols (pilot and comparative studies, S1 and S2 Appendices, respectively) and the statistical analysis plan (comparative study, S2 Appendix), which was finalised before the data were locked for analysis.

### Participants

Healthy adults were eligible for the study if they met all eligibility criteria (S1 and S2 Texts). Briefly, participants were malaria naïve, males (pilot study), or males and nonpregnant females (comparative study), aged 18–55 years. All participants gave written informed consent before enrollment and were compensated financially for their involvement in this study.

### Randomisation and masking

Randomisation schedules were generated using STATA release 13 (StataCorp LLC, www.stata.com). Participants in the comparative study were to be randomised in a 2:1 ratio to receive either ART-R or ART-S parasites. However, because of recruitment limitations, 3 participants in each cohort were randomised to receive ART-S parasites and the remainder randomised to receive ART-R parasites. The study was open label.

### Procedures

The ART-R *P. falciparum* isolate (Cam3.II$^{R539T}$, here referred to as the K13$^{R539T}$ strain) was originally collected from a patient in Cambodia with natural malaria infection [4]. The ART-R master cell bank was manufactured using a bioreactor system as previously described [10] and characterised at the time of release (S3 Text). Although the master cell bank was screened for blood-borne pathogens, the risk of pathogen transmission cannot be completely excluded with any blood transfusion, although the risk should be considerably lower than a single-unit blood transfusion in which the volume transfused is >5 logs higher. In vitro drug resistance testing and the ring-stage survival assay confirmed the ART-R parasites were resistant to artemisinin and sensitive to piperaquine (S3 Text and S1 Table). ART-R (K13$^{R539T}$ strain) and ART-S (3D7 strain) parasite inocula were produced using methods previously described [9].

In the pilot study, participants were intravenously inoculated with approximately 2,800 viable ART-R parasite-infected erythrocytes on Day 0. In the comparative study, participants were intravenously inoculated with approximately 2,800 viable ART-R parasite-infected erythrocytes on Day 0, or with approximately 2,800 viable ART-S parasite-infected erythrocytes on Day 1. ART-R infected participants were inoculated a day earlier than ART-S infected

participants so that parasites were at a similar stage of development at the time of AS administration (see Results section for more detail). Parasitaemia was monitored by quantitative PCR (qPCR) targeting the *P. falciparum* 18S rRNA gene [11]. Gametocytaemia was monitored by reverse transcription qPCR (qRT-PCR) for female-specific *pfs25* mRNA and male-specific *pfMGET* mRNA [12].

Single-dose oral AS (approximately 2 mg/kg; Guilin Pharmaceutical Shanghai Co. Ltd) was administered in tablet form on Day 9 in both studies. If recrudescence occurred after AS administration, participants received an oral dose of 120 mg dihydroartemisinin (DHA)/960 mg piperaquine phosphate (PQP, Eurartesim, Alfasigma S.p.A.; pilot study) or 960 mg PQP (PCI Pharma Services; comparative study). On approximately Day 26, or earlier if a second recrudescence occurred, all participants received a standard curative course, taken orally, of atovaquone/proguanil (A/P, Malarone, GlaxoSmithKline Australia Pty Ltd). Participants also received an oral dose of 45 mg primaquine (PQ, Primacin, Boucher & Muir Pty Ltd) to clear gametocytes at the end of the study (pilot study and comparative study Cohort 1) or on approximately Day 23 (comparative study cohorts 2 and 3). Transmissibility to mosquitoes was investigated in the comparative study as an exploratory objective; these results will be reported separately. Participants had regular follow-up visits until the End of Study visit on Day 90 ± 14 days (pilot study) or Day 28 ± 3 days (comparative study). Fig 1 illustrates the study design. The study procedures and timing of assessments are summarised in S2 and S3 Tables.

Development of parasitaemia was monitored daily from Day 3 (pilot study Participant 1) or Day 4 (pilot study Participant 2 and comparative study) until parasites were detected by qPCR, then twice daily until AS administration. Parasitaemia was measured post-AS at specified timepoints until 84 hours (S4 Table), then twice daily, daily, or every second day until parasitaemia was low and stable, then 3 times per week until A/P administration, and at the end of the study.

Plasma concentrations of AS and its active metabolite DHA were measured in blood samples (S4 Text) at the time points specified in the protocol.

## Outcomes

A primary outcome of both studies was safety and tolerability of infection with ART-R parasites (and of ART-S parasites in the comparative study only), determined by evaluating adverse events, physical examinations, vital signs, clinical biochemistry, haematology, and urinalysis. In the pilot study, another primary outcome was the infectivity of ART-R parasites, which was determined by the presence of parasites in inoculated participants as measured by qPCR. In the comparative study, another primary outcome was to compare parasite clearance profiles of ART-R and ART-S parasites after single-dose AS administration, which was determined by estimating the slope of parasite clearance curves, the parasite reduction ratio per 48 hours in the logarithmic scale ($\log_{10}PRR_{48}$), and parasite clearance half-lives. Based on previous IBSM studies with ART-S parasites, AS clearance data from ART-R strains in the field [2], and data from the pilot study, we hypothesised that mean parasite clearance after AS administration would be 30% slower for ART-R infected participants compared to ART-S infected participants [13]. Both studies had the exploratory outcome to calculate the following pharmacokinetic parameters for AS and DHA: area under the concentration-time curve from time 0 to the last measurable concentration ($AUC_{0-last}$); area under the concentration-time curve from time 0 to infinite time ($AUC_{0-\infty}$); maximum concentration ($C_{max}$); time of $C_{max}$ ($t_{max}$); and elimination half-life ($t_{1/2}$).

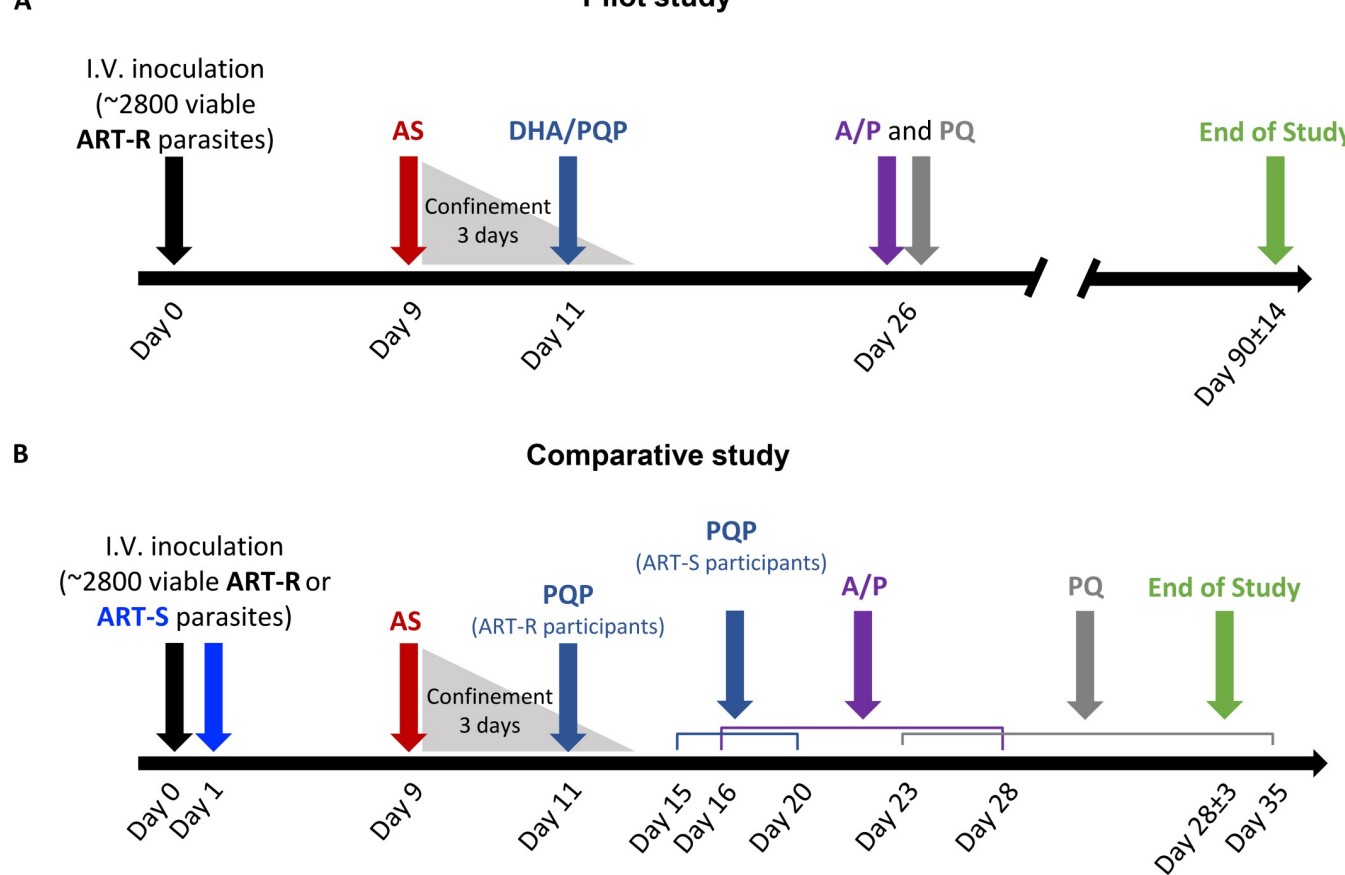

**Fig 1. Study design.** (A) In the pilot study, participants were IV inoculated with approximately 2,800 viable ART-R parasite-infected erythrocytes on Day 0, followed by AS administration on Day 9. Participants were confined at the clinical site for 3 days, during which they were administered DHA/PQP on Day 11. A/P and PQ administration occurred on Day 26 with the End of Study visit on Day 90 ± 14 days. (B) In the comparative study, participants were IV inoculated with approximately 2,800 viable ART-R parasite-infected erythrocytes (Day 0) or with approximately 2,800 viable ART-S parasite-infected erythrocytes (Day 1) and administered AS on Day 9. PQP administration occurred on Day 11 for ART-R infected participants and between Days 15 and 20 for ART-S infected participants. In the comparative study, the majority of participants were followed up after the End of Study visit for qPCR and/or safety monitoring. The timings of A/P administration, PQ administration, and the End of Study visit are also shown. ART-R, artemisinin-resistant; ART-S, artemisinin-sensitive; A/P, atovaquone/proguanil; AS, artesunate; DHA/PQP, dihydroartemisinin/piperaquine phosphate; IV, intravenously; PQ, primaquine; qPCR, quantitative PCR

## Statistical analysis

The sample size of the comparative study was calculated with nQuery Advisor version 7.0 (Statsols, www.statsols.com). Satterthwaite $t$ test of equal means but unequal variances was used to calculate sample size with a one-sided $\alpha = 0.05$, an estimated 30% mean difference in parasite clearance slope between ART-R and ART-S parasites, and a power of 80%. The estimated sample size using a 2:1 ratio of ART-R to ART-S was 18 to 9 participants. Participants were to be enrolled in 3 cohorts of 9 participants each. Review of parasitaemia data between cohorts was conducted using an adaptive design to re-evaluate the sample size necessary to reach the primary outcomes.

The parasite clearance slope is the slope of the linear regression of $\log_{10}$ parasitemia over time after AS administration, derived from an iterative procedure [13]. The weighted mean of the parasite clearance slope was estimated by the inverse-variance method using the corresponding standard errors of the clearance slope and presented as $\log_{10}PRR_{48}$ and the parasite clearance half-life, as previously described [13]. An omnibus test [14] was used to determine

differences between the weighted mean parasite clearance slopes of ART-R and ART-S parasites. Analyses were determined using R version 3.5.0.

Parasite growth dynamics were estimated using a sine-wave model to capture the oscillatory nature of the parasite growth [15]. The sine-wave growth model was applied to the pre-AS qPCR data using R version 3.5.0 and R package nlme version 3.1–137. For the pilot study, the sine-wave growth model was estimated for each participant, and the life cycle duration of ART-R parasites was estimated from the inverse-variance weighted mean of the sine-wave period estimates (S5 Text). For the comparative study, the parasite growth dynamics for ART-R and ART-S parasites were modelled using the sine-wave mixed-effects model (S5 Text).

Pharmacokinetic parameters were determined by noncompartmental analysis using R version 3.4.2 and R package IQRtools version 0.9.1 (IntiQuan GmbH). Mann–Whitney *U* tests were used to compare pharmacokinetic parameters of ART-R and ART-S infected participants.

## Results

The pilot study (*n* = 2) was conducted from April 27, 2017 to September 12, 2017, and the comparative study (*n* = 25) from October 26, 2017 to October 18, 2018 (Fig 2). In the comparative study, 25 participants were enrolled over 3 cohorts and randomised to receive ART-R (*n* = 16) ART-S (*n* = 9) parasites. Three participants randomised to ART-R parasites withdrew prior to inoculation, giving a final ratio of 13:9 ART-R to ART-S infected participants. No more participants were enrolled because the primary outcomes were met. All inoculated participants completed the study and were included in all final analyses. Participant baseline characteristics are summarised in Table 1.

In the pilot study, parasitaemia was detectable by qPCR on Day 4 and increased until AS administration on Day 9. Parasitaemia decreased in both participants post-AS administration until recrudescence occurred on Day 11, when DHA/PQP was administered. Parasitaemia did not completely clear after DHA/PQP administration (S1 Fig), with qRT-PCR indicating that persisting parasites included gametocytes (S5 Table). Parasites cleared completely after A/P and PQ administration on Day 26.

In the pilot study, the weighted mean parasite life cycle duration of ART-R parasites prior to AS administration was 46.7 hours (95% confidence interval [CI]: 43.8–49.6 hours; S2 Fig), which is longer than the approximately 39 hours previously estimated for ART-S parasites in IBSM studies [16]. Therefore, for the comparative study, ART-R infected participants were inoculated a day earlier than ART-S infected participants, so that parasites were at a similar stage of development in all participants at the time of AS administration.

In the comparative study, parasitaemia increased in both ART-R and ART-S infected participants until AS administration on Day 9 (Fig 3). Median parasitaemia prior to AS administration for ART-R infected participants was 71,427 parasites/mL (range 17,611–346,298 parasites/mL) and for ART-S infected participants 31,400 parasites/mL (range 10,625–393,678 parasites/mL). Parasite growth curves indicated that at the time of AS administration both ART-R and ART-S parasites were at a similar stage of development (S3 Fig).

In the comparative study, parasitaemia decreased in ART-R infected participants post-AS administration, with recrudescence occurring on Day 11, when PQP was administered (Fig 3). Parasitaemia in ART-S infected participants decreased post-AS administration to lower levels than ART-R infected participants (Fig 3). Persisting low levels of parasitaemia were observed in ART-S infected participants before recrudescence occurred between Days 11.5 and 16 (median Day 13); PQP was administered for recrudescence (Fig 3 and S6 Table).

Pilot Study

Comparative Study

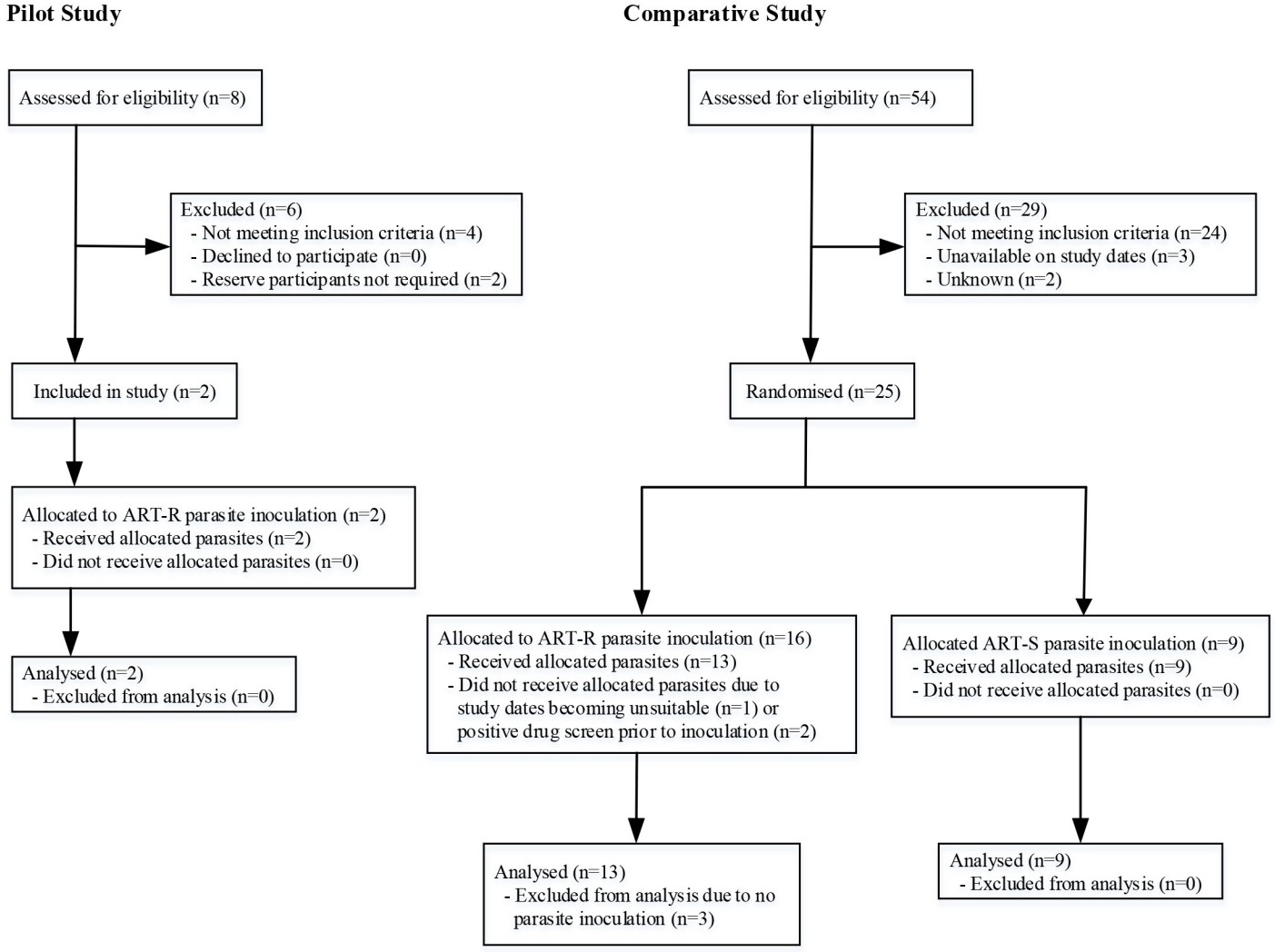

**Fig 2. Clinical trial profile.** In the pilot study, 2 participants were enrolled in 2 cohorts and inoculated with ART-R parasites, with a 3-week interval between cohorts (Cohort 1, $n = 1$; Cohort 2, $n = 1$). In the comparative study, 22 participants were inoculated with ART-R ($n = 13$) or ART-S ($n = 9$) parasites. Inoculated participants were enrolled in 3 cohorts: Cohort 1 (ART-R, $n = 3$; ART-S, $n = 3$), Cohort 2 (ART-R, $n = 7$; ART-S, $n = 3$), and Cohort 3 (ART-R, $n = 3$; ART-S, $n = 3$). ART-R, artemisinin-resistant; ART-S, artemisinin-sensitive.

Parasitaemia decreased in all ART-R and ART-S infected participants after PQP administration. Five ART-R infected participants had a second recrudescence of parasitaemia after PQP administration. qRT-PCR indicated that gametocytes were present in all participants (S5 Table). Following A/P and PQ administration (S6 Table), parasites were cleared completely in all participants (Fig 3).

In the pilot study, the parasite clearance half-life after AS administration was 5.4 hours (Table 2 and S7 Table). In the comparative study, parasite clearance half-life in ART-R infected participants after AS administration was twice as long than in ART-S infected participants (6.5 hours versus 3.2 hours; $p < 0.001$; Table 2 and S7 Table).

During the pilot study, 36 adverse events were reported in 2 participants, whereas 277 adverse events were reported in 22 participants in the comparative study (Table 3). No serious adverse events were reported. Most adverse events were mild and attributable to malaria. Common adverse events in both studies were headache, pyrexia, myalgia, nausea, and chills

**Table 1. Participant baseline characteristics.**

| Participant characteristics | | Pilot study | Comparative study | |
|---|---|---|---|---|
| | | Artemisinin-resistant ($N$ = 2) | Artemisinin-resistant ($N$ = 13) | Artemisinin-sensitive ($N$ = 9) |
| **Age (years)** | Median (range)[a] | 26 (23–28) | 22 (18–40) | 28 (22–35) |
| **Sex** | Male, $n$ (%) | 2 (100%) | 7 (53.8%) | 5 (55.6%) |
| | Female, $n$ (%) | NA | 6 (46.2%) | 4 (44.4%) |
| **Race** | White, $n$ (%) | 1 (50.0%) | 11 (84.6%) | 7 (77.8%) |
| | Asian, $n$ (%) | 1 (50.0%) | 0 (0.0%) | 2 (22.2%) |
| | Native Hawaiian or Other Pacific Islander, $n$ (%) | 0 (0.0%) | 1 (7.7%) | 0 (0.0%) |
| | Native South American, $n$ (%) | 0 (0.0%) | 1 (7.7%) | 0 (0.0%) |
| **Ethnicity** | Not Hispanic/Latino | NR | 12 (92.3%) | 9 (100%) |
| | Hispanic/Latino | NR | 1 (7.7%) | 0 (0.0%) |
| **Height (cm)** | Median (range)[a] | 181 (176–185) | 173 (161–187) | 172 (162–180) |
| **Weight (kg)** | Median (range)[a] | 74.5 (71.3–77.7) | 72.3 (59.4–103.8) | 77.9 (54.2–85.5) |
| **BMI (kg/m²)** | Median (range)[a] | 22.9 (22.7–23.0) | 25.3 (20.0–29.7) | 26.6 (19.6–31.4) |

Table shows only participants who were inoculated with malaria parasites. All data are values recorded at screening.

[a]For the pilot study, values are mean (range) because the median of 2 participants could not be determined.

**Abbreviations**: BMI, body mass index; NA, not applicable; NR, not recorded

(S8 Table). A total of 11 adverse events in ART-R infected participants ($n$ = 5) and 2 adverse events in ART-S infected participants ($n$ = 1) were related to antimalarial drugs (S9 Table).

Ten severe adverse events were recorded in 7 participants, all related to malaria. Four participants experienced transient falls in white cell counts that were classified as severe: transient leukopenia (ART-R $n$ = 1, $1.8 \times 10^9$/L, lower limit of normal [LLN] = $3.5 \times 10^9$/L); lymphopenia ART-R $n$ = 1, $0.49 \times 10^9$/L, LLN = $1 \times 10^9$/L); and neutropenia (ART-R $n$ = 2, $0.99 \times 10^9$/L, $0.95 \times 10^9$/L; ART-S $n$ = 1, $0.89 \times 10^9$/L; LLN = $1.5 \times 10^9$/L). A total of 2 ART-R infected participants and 2 ART-S infected participants had transient increases in alanine aminotransferase (ALT) levels that were classified as severe and peaked at 250 U/L ($6.3 \times$ upper limit of normal [ULN]), 248 U/L ($6.2 \times$ ULN), 247 U/L ($8.2 \times$ ULN), and 301 U/L ($10.0 \times$ ULN). One ART-S infected participant had an ALT elevation classified as severe concurrent with an elevation in aspartate aminotransferase that peaked at 225 U/L ($6.4 \times$ ULN), also classified as severe; no participant had a clinically significant increase in bilirubin.

In the comparative study, 1 ART-S infected participant and 1 ART-R infected participant developed ventricular extrasystoles; both began on Day 9 prior to AS administration. These ventricular extrasystoles were assessed as moderate in severity and possibly related to malaria. Serial electrocardiogram monitoring of the ART-S infected participant showed frequent isolated monomorphic ventricular ectopy, whereas the ART-R infected participant had ventricular bigeminy, trigeminy, and couplets. Serial measurement of troponin I levels (S10 Table) and echocardiography for both participants, as well as cardiac magnetic resonance imaging in the ART-R infected participant, ruled out structural cardiac abnormalities or myocardial damage. Follow-up by a specialist cardiologist of both participants indicated that the ventricular extrasystoles were attributable to unmasking of a predisposition to benign fever-induced tachyarrhythmia. Additionally, 4 ART-R infected participants and 1 ART-S infected participant were noted to have transient prolongation in their QT interval corrected using Fridericia's formula (QTcF) between ≥30 ms and ≤60 ms during the comparative study compared to either the pre-inoculum baseline and/or the pre-PQP baseline (including the ART-R infected participant with ventricular bigeminy; S11 Table). All adverse events, except the 2 cases of

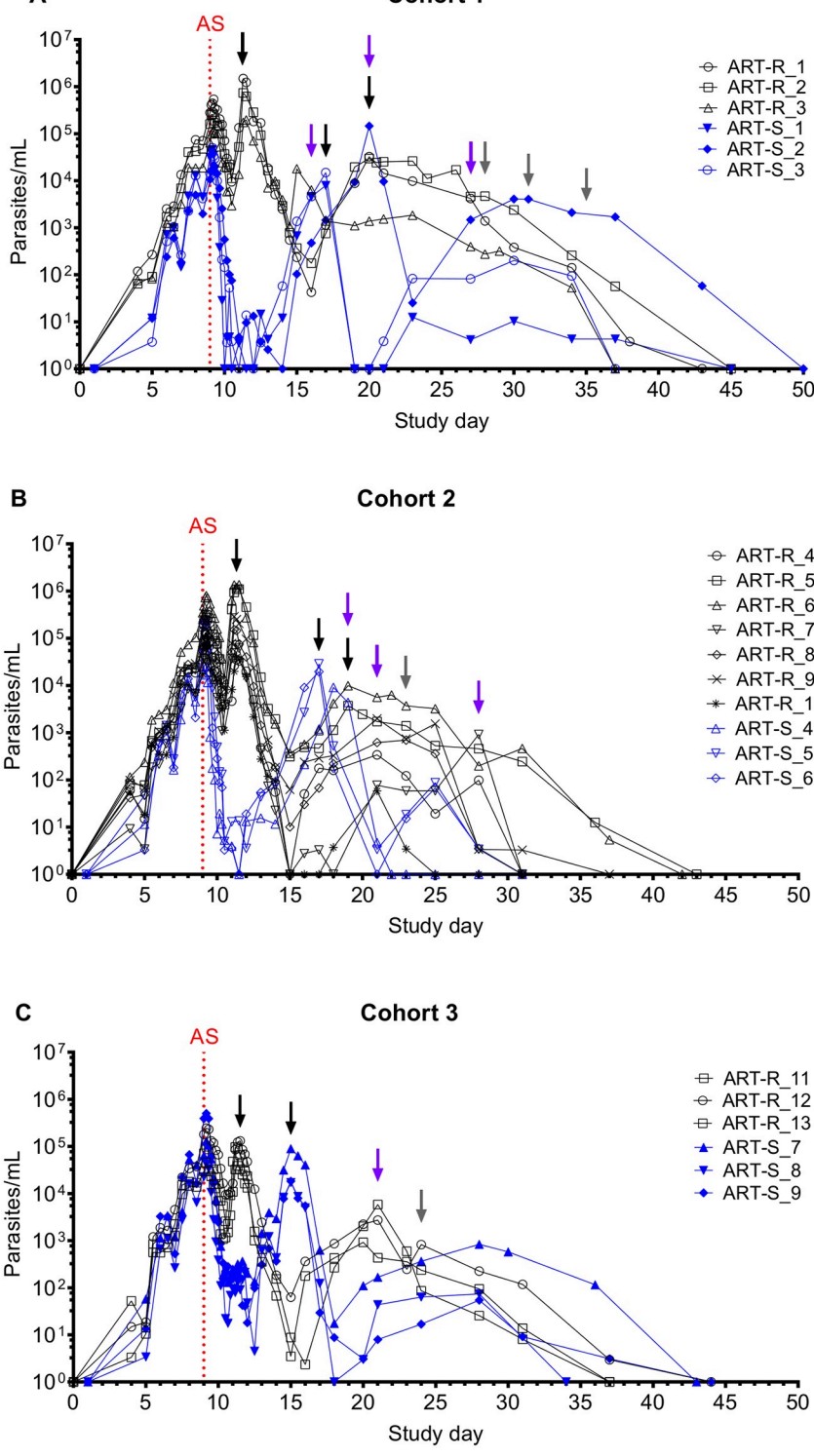

**Fig 3. Individual parasite growth and clearance profiles in the comparative study.** Parasitaemia of participants enrolled in Cohort 1 (A), Cohort 2 (B), and Cohort 3 (C) of the comparative study, as determined by 18S quantitative PCR. Participants were inoculated with parasites on Day 0 (ART-R) or Day 1 (ART-S) and administered AS on Day 9 (red dashed line). Black arrows represent PQP administration, and purple arrows represent A/P administration. All ART-R infected participants received PQP on Day 11, whereas ART-S infected participants received PQP between Days 15 and 20. PQ was also administered before the end of the study to clear gametocytes (grey arrows). A/P,

atovaquone/proguanil; ART-R, artemisinin-resistant; ART-S, artemisinin-sensitive; AS, artesunate; PQP, piperaquine; PQ, primaquine.

ventricular extrasystoles, resolved by the end of the study (pilot study, Day 90; comparative study, Day 55).

Pharmacokinetic parameters of AS and DHA for the pilot study are presented in S4 Fig and S12 Table and for the comparative study in S5 Fig and S13 Table. ART-R and ART-S infected participants in the comparative study showed similar pharmacokinetic profiles (S5 Fig).

## Discussion

In this study, we developed the first (to our knowledge) human model of ART-R *P. falciparum* malaria. Infection of healthy, malaria-naïve participants with ART-R parasites was safe and

**Table 2. Parasite clearance slope, $log_{10}PRR_{48}$, and parasite clearance half-life after AS administration.**

| Parasite clearance parameter | Pilot study | Comparative study | | p value[a] |
|---|---|---|---|---|
| | Artemisinin-resistant (N = 2) | Artemisinin-resistant (N = 13) | Artemisinin-sensitive (N = 9) | |
| Parasite clearance slope (95% CI) | −0.056 (−0.062 to −0.051) | −0.046 (−0.048 to −0.045) | −0.096 (−0.100 to −0.091) | <0.001 |
| $Log_{10}PRR_{48}$ (95% CI) | 2.70 (2.43–2.97) | 2.23 (2.15–2.31) | 4.59 (4.38–4.79) | <0.001 |
| Parasite clearance half-life [hours] (95% CI) | 5.4 (4.9–6.0) | 6.5 (6.3–6.7) | 3.2 (3.0–3.3) | <0.001 |

$Log_{10}PRR_{48}$ and parasite clearance half-life are transformations of the weighted mean parasite clearance slope.

[a]p value is for the comparison of artemisinin-resistant and artemisinin-sensitive infected participants in the comparative study. p value was obtained using an omnibus test on weighted mean clearance slope estimates of artemisinin-resistant and artemisinin-sensitive parasites.

**Abbreviations**: AS, artesunate; CI, confidence interval; $log_{10}PRR_{48}$, parasite reduction ratio per 48 hours in the logarithmic base 10 scale

**Table 3. Adverse events reported during the studies.**

| Adverse events category | Pilot study | Comparative study | |
|---|---|---|---|
| | Artemisinin-resistant (N = 2) | Artemisinin-resistant (N = 13) | Artemisinin-sensitive (N = 9) |
| | | n (%) M | |
| **Participants with at least one adverse event** | 2 (100%) 36 | 13 (100%) 151 | 9 (100%) 126 |
| **Participants with at least one adverse event by maximum severity[a]** | | | |
| Mild | 0 (0.0%) 23 | 1 (7.7%) 116 | 1 (11.1%) 92 |
| Moderate | 1 (50.0%) 12 | 9 (69.2%) 30 | 5 (55.6%) 30 |
| Severe | 1 (50.0%) 1 | 3 (23.1%) 5 | 3 (33.3%) 4 |
| Participants with at least one adverse event related to malaria | 2 (100%) 29 | 13 (100%) 117 | 9 (100%) 108 |
| Participants with at least one adverse event related to study drugs | 1 (50.0%) 4 | 4 (30.8%) 7 | 1 (11.1%) 2 |
| Participants with at least one adverse event related to mosquito feeding[b] | NA | 6 (46.2%) 6 | 1 (11.1%) 1 |
| Participants with a serious adverse event | 0 (0.0%) 0 | 0 (0.0%) 0 | 0 (0.0%) 0 |
| Participants with an adverse event leading to participant discontinuation | 0 (0.0%) 0 | 0 (0.0%) 0 | 0 (0.0%) 0 |

The table shows only participants who were inoculated with malaria parasites and completed the study.

[a]Number of occurrences (M) of adverse events in this row is not by maximum severity, but total mild, moderate, and severe adverse events.

[b]Mosquito feeding was not performed in the pilot study and was performed in all artemisinin-resistant infected participants and one out of 9 artemisinin-sensitive infected participants in the comparative study.

**Abbreviations**: M, number of occurrences of adverse events; NA, not applicable

well tolerated. ART-R parasites cleared from all participants with available antimalarial drugs. As expected, ART-R parasites took longer to clear than ART-S parasites.

The observed difference in parasite clearance between ART-R infected participants and ART-S infected participants was larger than the 30% difference in mean parasite clearance rates that the study was powered to detect. The longer parasite clearance half-life after a single-dose of AS observed in ART-R infected participants compared with ART-S infected participants (6.5 versus 3.2 hours) agrees with the values reported in malaria patients treated with AS in the Greater Mekong subregion (>5 hours) [1–3,17]. These results are in contrast to a recently reported study of splenectomised *Aotus* monkeys infected with either wild-type or ART-R mutant parasites in which no difference in parasite clearance half-life was reported [18].

Parasite clearance after drug treatment is a complex process involving factors such as the mechanism of action of the drug, splenic clearance of erythrocytes containing dead or damaged parasites, and the effect of host immunity. The delayed clearance observed in field ART-R infections could be confounded by these factors [19]. However, our studies used malaria-naïve participants given a single dose of antimalarial drug rules out these confounding effects.

In the comparative study, we inoculated ART-R infected participants a day earlier than ART-S infected participants to account for the longer duration of the parasite life cycle of ART-R parasites, as estimated with the data from the pilot study. By inoculating ART-R infected participants one day earlier, we aimed to administer AS at the same stage of parasite life cycle for both strains. Therefore, the difference in life cycle duration between the 2 strains is not expected to influence the comparison of parasite clearance.

This study was designed to characterise the pharmacodynamic response after a single dose of AS; therefore, we anticipated that recrudescence would occur [20]. The persisting low parasitaemia observed in ART-S infected participants after AS administration could indicate the presence of dormant parasites, which have been previously described in vitro [21,22], or could be a consequence of a fraction of ART-S parasites surviving single-dose AS. Further investigation is required to understand the reason for this persisting low parasitaemia. The recrudescence after PQP administration (960 mg) that occurred in 5 ART-R infected participants was likely due to insufficient drug exposure to kill all parasites rather than to drug resistance because the ART-R parasite strain tested PQP-sensitive by in vitro assay.

The adverse events observed in this study were in line with previous IBSM studies with ART-S parasites conducted at our site and elsewhere [6–9]. The transaminase elevations were transient and asymptomatic and were not associated with clinically significant elevations in bilirubin. Similar liver function changes have been reported in previous IBSM studies [8] and sporozoite challenge studies [23], as well as in natural malaria infections [24,25], indicating these elevations are likely due to the inflammatory process in malaria infection.

No evidence of myocardial damage was detected as a cause of the 2 cases of ventricular extrasystoles that occurred during the comparative study. After comprehensive investigations, both adverse events were attributed to undiagnosed predispositions to cardiac arrhythmias that were unmasked by the febrile illness. Cardiac events with associated myocardial damage have been reported in malaria VIS with sporozoites [26,27]. Furthermore, serendipitous detection of benign cardiac arrhythmias has been documented in early phase clinical trials [28,29].

The main limitation of this study is that the ART-R and ART-S parasite strains do not share the same genetic background. An optimal strain for comparison may have been the ART-S parasite generated by back mutation of the ART-R strain [4]. Furthermore, the R539T genotype has been outcompeted by other genotypes such as C580Y [30]. However, observations from field studies have demonstrated that parasite clearance half-life in patients carrying either

the R539T or the C580Y mutation are similar [31]. Therefore, we believe that our model is suitable to investigate drugs effective against ART-R parasites. Furthermore, both the R539T and the C580Y strains are widely distributed across the Greater Mekong subregion [31,32]. Another limitation of the study is that our results are difficult to generalize to other populations. All the participants in these studies were malaria naïve, and therefore their response to malaria infection may be different to patients in endemic areas, where patients may have different levels of immunity to malaria. In addition, levels of parasitaemia are lower in VIS than in clinical malaria, which further hinders generalisability. Nevertheless, despite these limitations, the delayed clearance profile of ART-R parasites after AS observed in these studies is comparable to observations from field studies [1–3,17].

This is the first time that a *Plasmodium* spp. strain has been isolated from a patient with clinical malaria, used to manufacture a master cell bank in vitro in compliance with good manufacturing practice, and inoculated into healthy participants. We have shown that this approach is safe and can be used for VIS with other parasite strains for drug development.

In conclusion, we developed the first (to our knowledge) human model of ART-R *P. falciparum* malaria. Although based on a relatively small sample size, results indicate that this ART-R IBSM model is safe and replicates the delayed parasite clearance phenotype documented in the field. The ART-R IBSM model will be a powerful tool to test the efficacy of candidate antimalarial drugs against ART-R infections.

## Supporting information

**S1 Text. Eligibility criteria for the pilot study.**
(PDF)

**S2 Text. Eligibility criteria for the comparative study.**
(PDF)

**S3 Text. Summary testing of the artemisinin-resistant (K13$^{R539T}$) *P. falciparum* master cell bank.**
(PDF)

**S4 Text. Determination of AS and DHA concentrations in plasma samples.** AS, artesunate; DHA, dihydroartemisinin.
(PDF)

**S5 Text. Parasite growth dynamics.**
(PDF)

**S1 Fig. Individual parasite growth and clearance profiles in the pilot study.**
(PDF)

**S2 Fig. Parasite life cycle duration of artemisinin-resistant parasites in the pilot study.**
(PDF)

**S3 Fig. Sine-wave growth model for artemisinin-resistant and artemisinin-sensitive parasites in the comparative study.**
(PDF)

**S4 Fig. Plasma concentrations of AS and DHA in the pilot study.** AS, artesunate; DHA, dihydroartemisinin.
(PDF)

**S5 Fig. Geometric mean dose-normalised plasma concentration of AS and DHA in the comparative study.** AS, artesunate; DHA, dihydroartemisinin.
(PDF)

**S1 Table. In vitro antimalarial drug resistance testing of the artemisinin-resistant (K13$^{R539T}$)** *P. falciparum* **master cell bank.**
(PDF)

**S2 Table. Schedule of events for the pilot study.**
(PDF)

**S3 Table. Schedule of events for the comparative study.**
(PDF)

**S4 Table. Malaria 18S qPCR timepoints from pre-AS to 84 hours post-AS administration.** AS, artesunate; qPCR, quantitative PCR.
(PDF)

**S5 Table. Gametocytaemia prior to PQ administration.** PQ, primaquine.
(PDF)

**S6 Table. Antimalarial administration days.**
(PDF)

**S7 Table. Individual parasite clearance slope, log$_{10}$PRR$_{48}$, and parasite clearance half-life after AS administration.** AS, artesunate.
(PDF)

**S8 Table. Adverse events by System Organ Class, Preferred Term and** *P. falciparum* **strain.**
(PDF)

**S9 Table. Summary of adverse events related to antimalarial drugs.**
(PDF)

**S10 Table. Troponin I results for participants in the comparative study with ventricular extrasystoles.**
(PDF)

**S11 Table. Summary of QTcF prolongations $\geq$30 ms in the comparative study.** QTcF, QT interval corrected using Fridericia's formula.
(PDF)

**S12 Table. Pharmacokinetic parameters of AS and DHA in the pilot study.** AS, artesunate; DHA, dihydroartemisinin.
(PDF)

**S13 Table. Pharmacokinetic parameters of AS and DHA in the comparative study.** AS, artesunate; DHA, dihydroartemisinin.
(PDF)

**S1 Checklist. CONSORT checklist.**
(PDF)

**S1 Appendix. Protocol and raw data listings for the pilot study.**
(PDF)

**S2 Appendix. Protocol and raw data listings for the comparative study.**
(PDF)

## Acknowledgments

The authors thank all staff from Q-Pharm Pty Ltd, particularly Paul Griffin, Paul Morgan, Sue Mathison, Mark Armstrong, Anna Brischetto, and Miranda Goodwin; G. Dennis Shanks from the Australian Army Malaria Institute for serving as medical monitor; all QIMR Berghofer Clinical Tropical Medicine staff, particularly Helen Jennings, Rebecca Pawliw, Rebecca Farrow, Katharine Trenholme, and Bridget Barber; staff of the Queensland Paediatric Infectious Diseases Laboratory for performing qPCR analysis; staff of Pathology Queensland, particularly Brett McWhinney for drug concentration levels; Peter O'Rourke and Stacey Llewellyn for statistical advice; the Australian Red Cross Blood Service for providing human erythrocytes for the master cell bank production; Will Parsonage for cardiac investigations; Rick Fairhurst and Leann Tilley for providing the artemisinin-resistant strain; and David Fidock for providing the artemisinin-resistant strain and for critical review of the manuscript. The authors also thank the volunteers who participated in the studies.

## Author Contributions

**Conceptualization:** Rebecca E. Watts, Anand Odedra, Stephan Chalon, Jörg J. Möhrle, James S. McCarthy.

**Formal analysis:** Rebecca E. Watts, Louise Marquart, Lachlan Webb, Azrin N. Abd-Rahman, Maria Rebelo, Zuleima Pava.

**Funding acquisition:** Stephan Chalon, Jörg J. Möhrle, James S. McCarthy.

**Investigation:** Anand Odedra, Cielo Pasay, Nanhua Chen, Christopher L. Peatey, James S. McCarthy.

**Methodology:** Rebecca E. Watts, Anand Odedra, Louise Marquart, Laura Cascales, Stephan Chalon, Maria Rebelo, Zuleima Pava, Katharine A. Collins, Cielo Pasay, Nanhua Chen, Christopher L. Peatey, Jörg J. Möhrle, James S. McCarthy.

**Project administration:** Rebecca E. Watts, Anand Odedra, Laura Cascales.

**Supervision:** Rebecca E. Watts, Anand Odedra, Stephan Chalon, Jörg J. Möhrle, James S. McCarthy.

**Visualization:** Rebecca E. Watts, Laura Cascales, Maria Rebelo, Zuleima Pava, James S. McCarthy.

**Writing – original draft:** Rebecca E. Watts, Laura Cascales, Maria Rebelo, Zuleima Pava, James S. McCarthy.

**Writing – review & editing:** Rebecca E. Watts, Anand Odedra, Louise Marquart, Lachlan Webb, Azrin N. Abd-Rahman, Laura Cascales, Stephan Chalon, Maria Rebelo, Zuleima Pava, Katharine A. Collins, Cielo Pasay, Nanhua Chen, Christopher L. Peatey, Jörg J. Möhrle, James S. McCarthy.

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
