## [Editor Report · Decision Letter 0]

10 Feb 2020

Dear Dr McCarthy, 

Thank you for submitting your manuscript entitled "Safety and parasite clearance of artemisinin-resistant Plasmodium falciparum infection: A randomised volunteer infection study" for consideration by PLOS Medicine.

Your manuscript has now been evaluated by the PLOS Medicine editorial staff and I am writing to let you know that we would like to send your submission out for external peer review.

Kind regards,

Helen Howard, for Clare Stone PhD 

Acting Editor-in-Chief

PLOS Medicine 

plosmedicine.org

---

## [Decision Letter · Decision Letter 1]

2 Apr 2020

Dear Dr. McCarthy,

Thank you very much for submitting your manuscript "Safety and parasite clearance of artemisinin-resistant Plasmodium falciparum infection: A randomised volunteer infection study" (PMEDICINE-D-20-00295R1) for consideration at PLOS Medicine. 

[LINK]

In light of these reviews, I am afraid that we will not be able to accept the manuscript for publication in the journal in its current form, but we would like to consider a revised version that addresses the reviewers' and editors' comments. Obviously we cannot make any decision about publication until we have seen the revised manuscript and your response, and we plan to seek re-review by one or more of the reviewers. 

We expect to receive your revised manuscript by Apr 23 2020 11:59PM. Please email us (plosmedicine@plos.org) if you have any questions or concerns.

We look forward to receiving your revised manuscript. 

Sincerely,

Clare Stone, PhD

Managing Editor 

PLOS Medicine

plosmedicine.org

Academic Editor comments:

This is a descriptive study but readers may be helped if the authors could formulate a clear research question which could be presented at the end of the introduction?

Editor comments: 

Title: Please revise your title according to PLOS Medicine's style. Would it be more accurate to call your study 2 phase 1 trials? Please add the country setting.

Abstract – Please add summary demographic information including mean age; Please add information regarding country setting and towns and recruitment.

Main text – Please ensure refs are in square brackets before punctuation.

Data – note an author cannot be a point of contact for data requests. Please provide another contact such as a data manager. 

Throughout the text there is a lack of information about the locations of these trials as well as where recruitment was. Please provide. 

Please use sections and paragraphs in the CONSORT as page numbers can change during revision and formatting. 

Did your study have a prospective protocol or analysis plan? Please state this (either way) early in the Methods section.

Data – you say some restrictions will apply, then say all data is available in the MS and Supp Files. Please state what data is available and what is not and why. 

Abstract – this should be formatted with 3 sections: Background, Method and Findings, Conclusion. Please format accordingly. 

Abstract – use months as well as years for recruitment and tell us the cities / regions where recruitment took place, if known; Please use 95% Cis as well as p values for all quantifiable data (both here and throughout the manuscript); Please ensure summary demographic information is provided, including gender ratio and mean age and number of smokers, etc; Please provide – as the last sentence of the ‘Methods and Findings’ section a sentence on the limitations of the study; 

Refs in main text, these need to be presented in square brackets and not using superscript. 

In the main text where you use 95%Cis, please also add p values. 

Please include line numbers on your revised version.

Please be careful of deriving cause where non can be drawn as this is not a trial: “Our data provide evidence…” and “In summary, our study provides evidence that obesity, even without metabolic syndrome…”

Please use the "Vancouver" style for reference formatting, and see our website for other reference guidelines https://journals.plos.org/plosmedicine/s/submission-guidelines#loc-references (noting remove ital font specifically)

Did your study have a prospective protocol or analysis plan? Please state this (either way) early in the Methods section.

Please ensure that the study is reported according to the STROBE guideline, and include the completed STROBE checklist as Supporting Information. Please add the following statement, or similar, to the Methods: "This study is reported as per the Strengthening the Reporting of Observational Studies in Epidemiology (STROBE) guideline (S1 Checklist)."

Comments from the reviewers:

Reviewer #1: To the authors, 

The manuscript "Safety and parasite clearance of artemisinin-resistant Plasmodium falciparum infection: a randomised volunteer infection study" by Watts et al describes with full details the establishment of an induced blood stage malaria model in human volunteers and the results they obtain in two assays.

The manuscript is well written, the study is nicely designed and incorporates all the necessary prerequisites for the selection of the volunteers, the monitoring of the infections and the clinical management of the patient.

Some minor amendments / precisions could give some valuable points:

- A justification of why from the beginning IBSM was preferred to sporozoites induced infections that are more close to natural ones? 

- While assessing parasitaemia by 18S qPCR is very precise, a justification of why not making blood film concurrently could be useful particularly to correlate with stages present and their development.

- The use of the name "K13 parasite" all along the study sounds inadequate and could be misleading since all P. falciparum (including the 3D7 use for comparison) harbour the K13 gene. The use another name or the real strain or isolate name could be preferable.

- The production of the master cell bank references the 3D7 strain. No precision is given for the production of ART-R parasite MCB and the study only mentioned as a limitation that both parasites have different genetic background. Was the same method (GAP) applied to the production of the MCB for the ART-R parasite? Which gene have been targeted? Could the authors give precision about ART-R parasite MCB such as parasite stages also represented by 96% of rings, presence of schizonts and gametocytes, parasitaemia?

- The ART-R strain used was R539T while the authors acknowledged that this strain have been outcompeted by some other such as the C580Y, they could also highlight that this strain is widely distributed in SEA with adequate reference.

- The listed serological tests include: HBs Ag, anti-HBc Ab, anti-HCV Ab, anti-HIV1 and anti-HIV2 Ab… Despite stringent selection criteria, did any serological tests for Malaria were performed?

- Gametocytaemia for both macro- and microgametocytes was tested by stage specific qRT-PCR and the results sounds promising for transmission experiments. Transmission experiments to mosquitoes were attempted as exploratory study but only very limited information is provided, and as stated "the results will be shared separately… With no need to provide the full details and results about these experiments it is highly desirable to state if the transmission was achieved or not, this will help validate the model presented.

- It would be good to present, in table format, the clinical biochemistry and haematology parameters recorded along the experiments with reference values and highlighting the abnormal ones. 

Overall, it is an impressive study and an interesting manuscript that contribute to the development of safe and useful human malaria infection models. Thank you.

Reviewer #2: In the proposed document, the authors report the results of a human challenge experiment. I have restricted my comments to the statistical analyses. The methods require additional detail.

The authors calculate 'The weighted mean of the parasite clearance slope', however, it is not clear what the parasite clearance slope is and what the weights are. This should be clarified.

The authors refer to an 'omnibus test', however no reference is made. A brief description of this test should be provided to help the non-specialist reader.

The authors conduct a 'sine-wave growth model' to model the viral titers (I believe). The authors should set out why this model was chosen (I presume for the sinusoidal nature of the dynamics). It would also be useful to demonstrate the fit of their model (e.g., plot the observed versus fitted average values).

The authors should clarify what the weights are for the sine-wave growth model.

Similarly, the authors state that 'Pharmacokinetic parameters were determined by non-compartmental analysis, however, it is not clear what a non-compartmental analysis represents. The authors should describe what they mean by this and clarify exactly what parameters are being estimated.

Reviewer #3: 

This is an impressive study of great importance. Human model for artemisinin resistant malaria could be valuable for a variety of reasons, including (but not limited to) controlled studies of new antimalarials. I have a few small comments, questions, and clarifications that should be addressed.

Major critique:

The limitations should be more comprehensively considered and explained. As I've already stated, this is an impressive amount of work. Pointing out potential limitations to this work does not detract from the quality of work. 

Currently, the only limitation that the authors discuss is related to the different genetic background between 3d7 and the k13 strain that they chose for this study. No explanation is given for why they chose the strain that they did. No comments are given about testing to see if their hypothesis (that they would have the same results if using a different strain) is true. 

No discussion is given about sample sizes or generalizability/representativity of the study population to other populations. 

On page 19: "However, our studies used malaria naïve participants given a single dose of antimalarial drug, which rules out these confounding effects." Some immunity is inherited rather than acquired, and immunity to malaria in general is not well understood. Were there any tests for blood cell variations among participants? Given that immunity is complex, and not perfectly understood, it is a bit too strong of a statement to suggest that having malaria naïve participants completely accounts for any possible confounding from immunity. Also, the inclusion criteria appears to limit potential malaria exposure to the last year. While some types of malaria immunity are short lived, immunological markers for malaria (msp, csp, etc.) can last for a very long time (years and possibly decades).

Minor critiques:

Title: Why is the term "volunteer infection study" used rather than "human challenge study"? Are they not the same? The latter is more commonly used and would be more clearly understood by most infectious disease specialists. 

Introduction: "This drug resistant phenotype is linked to specific mutations…" It would be a bit more accurate to describe this as mutations in a specific genome region, rather than specific mutations, since there are several different mutations in the kelch region that are associated with the phenotype of delayed parasite clearance. 

In the final paragraph of page 4, it would help if there was a sentence explaining that both the pilot and the comparative (second) study are being presented here. It becomes clear in the methods and results sections, but it would be good to state up front that this manuscript presents the results from both a pilot study and a follow-up study. A simple: "Here we present results from… " type statement here would work.

Why was the R539T mutation chosen? There are others of higher concern…

Page 6: It is a bit confusing to have the K13 inoculants on day 0 and 3d7 inoculants on day 1. Later you give a brief explanation for why, but it would add some clarity if it was explained here. 

Page 9: The pharmacokinetic parameters for artesunate and DHA should be explained. Some readers will not immediately recognize the abbreviations.

Page 12: Regarding the difference in parasite life cycle duration, which led to starting the K13 infected participants a day earlier… this deserves at least a statement or two in the conclusion. Also, if life stages are normally at different lengths, does this not influence the comparison of the 3d7 and the K13 resistant parasitemias? Or is this accounted for with the use of the mixed effects model? Please explain.

Page 12: Any explanation for the massive difference in parasitemia between the K13 lineage and 3d7 (with parasitemia over 2 times larger)?

Page 14: This isn't a minor point, but consider giving a quantitative value rather than stating that clearance was "significantly longer" - for example, it looks like clearance was twice as long for the k13 resistant infected participants when compared to the 3d7 infected participants.

[LINK]

---

## [Decision Letter · Decision Letter 2]

3 Jun 2020

Dear Dr. McCarthy,

Thank you very much for re-submitting your manuscript "Safety and parasite clearance of artemisinin-resistant Plasmodium falciparum infection: A pilot and a randomised volunteer infection study in Australia" (PMEDICINE-D-20-00295R2) for review by PLOS Medicine.

I have discussed the paper with my colleagues and the academic editor and it was also seen again by the original reviewers. I am pleased to say that provided the remaining editorial and production issues are dealt with we are planning to accept the paper for publication in the journal.

[LINK]

We look forward to receiving the revised manuscript by Jun 10 2020 11:59PM. 

Sincerely,

Clare Stone, PhD

Managing Editor 

PLOS Medicine

plosmedicine.org

Requests from Editors:

In the attached protocol documents, some of the authors are listed as having positions with "Q Pharm", which is apparently a company (CRO). If they have been paid by Q Pharm, I think this should be included in the competing interest statement. 

The data statement needs to be updated to note how the authors will make the aggregate data available; also please remove the author contact as this is not permitted.

Line 51 an elsewhere please add in to our knowledge re "the first" 

Line 146 – can you clarify what you mean by ‘which was finalised before the data was locked for analysis.’

Please remove all trademark signs

It seems from the protocol that the volunteers received money; Please mention this in the methods section

If I understand correctly the authors cannot rule out a small risk of blood-borne pathogen transmission in this system, and they could mention this in the methods section.

- At line 379, we learn that 7 participants had "severe adverse events related to malaria". This seems to have been missed in the abstract and author summary which note that there were "no serious adverse events". Please amend.

Please ensure the abstract is accurate on safety events. Please quote the severe adverse events up-front (including the 2 cardiac events, incidentally) and noting that these were not clinically serious (in their assessment)

Where the authors claim that the method can be "safely used", they can perhaps modify this by acknowledging the small size of their study

Please adjust statistics to style: p<0.0001 -> p<0.001 in abstract and at least one table

In the revised doc, when resubmitting, please add the marked up version as a Supp file instead of including with the clean doc in the main file. 

Comments from Reviewers:

Reviewer #1: Dear authors and editors,

The revision of the manuscript now titled "Safety and parasite clearance of artemisinin-resistant Plasmodium falciparum infection: A pilot and a randomised volunteer infection study in Australia", presents an impressive piece of work based on a large amount of data and detailed studies.

Encompassing, detailed answers to the reviewers' comments and the editorial requests, this revision has definitely gains in clarity, precision and readability. 

A single minor comment: since the study is based in Australia in the "theoretical risk zone" for autochthonous Babesia microti, did serology for Babesia was performed in the volunteers accounting that Babesia is often asymptomatic and some cross-protective immunity exists with Plasmodium? This could be mentioned.

Overall, this a convincing study presenting a new and valuable tool in the field of Plasmodium falciparum artemisinin-resistance. This tool will definitely be useful for further studies and has the potential to accelerate the development and evaluation of new anti-malarial drugs efficient against ART-R parasites which urgently needed. I will be happy to it published.

Reviewer #2: I have no further comments

Reviewer #3: The authors have satisfactorily addressed all of my questions and concerns.

[LINK]

---

## [Editor Report · Decision Letter 3]

21 Jul 2020

Dear Prof. McCarthy, 

On behalf of my colleagues and the academic editor, Dr. Lorenz von Seidlein, I am delighted to inform you that your manuscript entitled "Safety and parasite clearance of artemisinin-resistant Plasmodium falciparum infection: A pilot and a randomised volunteer infection study in Australia" (PMEDICINE-D-20-00295R3) has been accepted for publication in PLOS Medicine. 

PRODUCTION PROCESS

PRESS

PROFILE INFORMATION

Thank you again for submitting the manuscript to PLOS Medicine. We look forward to publishing it. 

Best wishes, 

Clare Stone, PhD

Managing Editor 

PLOS Medicine

plosmedicine.org